# Nutrient Density as a Dimension of Dietary Quality: Findings of the Nutrient Density Approach in a Multi-Center Evaluation

**DOI:** 10.3390/nu13114016

**Published:** 2021-11-10

**Authors:** Marieke Vossenaar, Noel W. Solomons, Siti Muslimatun, Mieke Faber, Olga P. García, Eva Monterrosa, Kesso Gabrielle van Zutphen, Klaus Kraemer

**Affiliations:** 1Center for Studies of Sensory Impairment, Aging and Metabolism (CeSSIAM), Guatemala City 01011, Guatemala; cessiam@cessiam.org.gt; 2Food Science and Nutrition Department, Indonesia International Institute for Life Sciences (i3L), Jakarta 13210, Indonesia; siti.muslimatun@i3l.ac.id; 3Non-Communicable Diseases Research Unit, South African Medical Research Council, Cape Town 7505, South Africa; mieke.faber@mrc.ac.za; 4Centre of Excellence for Nutrition, North-West University, Potchefstroom 2531, South Africa; 5School of Natural Sciences, Universidad Autónoma de Querétaro, Querétaro 76017, Mexico; olga.garcia@uaq.mx; 6Sight and Life, Kaiseraugst, 4002 Basel, Switzerland; emonterrosa@gainhealth.org (E.M.); kesso.vanzutphen@sightandlife.org (K.G.v.Z.); klaus.kraemer@sightandlife.org (K.K.)

**Keywords:** micronutrients, nutritious diets, nutrient density, energy expenditure, problem nutrients, women, obesity, Indonesia, Mexico, South Africa

## Abstract

The nutrient adequacy of a diet is typically assessed by comparing estimated nutrient intakes with established average nutrient requirements; this approach does not consider total energy consumed. In this multinational survey investigation in Indonesia, Mexico, and South Africa, we explore the applications of the “critical nutrient-density approach”—which brings energy requirements into the equation—in the context of public health epidemiology. We conducted 24 h dietary recalls in convenience samples of normal-weight (BMI 18.5–25 kg/m^2^) or obese (BMI > 30 kg/m^2^), low-income women in three settings (*n* = 290). Dietary adequacy was assessed both in absolute terms and using the nutrient density approach. No significant differences in energy and nutrient intakes were observed between normal-weight and obese women within any of the three samples (*p* > 0.05). Both the cut-point method (% of EAR) and critical nutrient density approach revealed a high probability of inadequate intakes for several micronutrients but with poor concordance between the two methods. We conclude that it may often require some approximate estimate of the habitual energy intake from an empirical source to apply a true critical nutrient density reference for a population or subgroup. This will logically signify that there would be more “problem nutrients” in the diets examined with this nutrient density approach, and efforts toward improved food selection or food- or biofortification will frequently be indicated.

## 1. Introduction

The nutrient adequacy of a diet is typically assessed by comparing estimated nutrient intakes with established average nutrient requirements. Because recommended intakes for micronutrients for any given age, sex, or physiological group do not vary by energy requirements, the total energy consumed is not taken into consideration. Nutrient density, defined as the ratio of nutrient to energy, brings energy into the equation.

One can expand the nutrient density concept to create a reference nutrient density value, referred to as the “critical nutrient density”, as described in our previous publication [1]. This is a reference value that consists of a recommended daily intake for a specific nutrient as the numerator and daily energy requirements as the denominator. When energy requirements are met, a diet with an adequate critical nutrient density is assured of meeting nutrient requirements. The gap between the nutrient density of the estimated diet and the critical nutrient density can be used to identify “problem nutrients”. A problem nutrient is one that is deficient when an individual is considered to be meeting his or her daily energy requirements.

The critical nutrient density of the diet of normal-weight versus obese women differs because their daily energy requirements differ (the denominator of the critical nutrient density equation is greater), even though they have the same recommended intakes for micronutrients (the numerator of the critical nutrient density equation is the same). Diets of obese women will have lower estimated critical nutrient densities than those of normal-weight women, giving them a greater energy “allowance” to meet nutrient requirements.

The application of nutrient density as a reference for adequate nutrient intakes in relation to energy requirements dates back to 1998 in the conceptualization of “desirable nutrient density” by Brown, Dewey, and Allen in the context of complementary feeding of infants and young children [2]. In this work, reference energy recommendations were used as the denominator in the density expression. The term “desirable nutrient density” was modified to “critical nutrient density” in the publications of Vossenaar and Solomons [3,4,5]. In the current work, this principle is used to address adult circumstances for the first time in our publication history as a departure from previous work in children.

In this multinational survey investigation [1,6], executed in three diverse regions, we explore the applications of the nutrient density and critical nutrient density approaches in public health epidemiology. The aim is to assess the dietary adequacy of the diet of normal-weight (body mass index (BMI) 18.5–25 kg/m^2^) and obese (BMI > 30 kg/m^2^) women in each country setting and to compare diets between these two target groups within each country setting, as well as for the pooled sample. Dietary adequacy is assessed both in absolute terms (i.e., estimated intakes of nutrients) and using the nutrient density approach (i.e., nutrient in relation to calorie intake).

## 2. Materials and Methods

### 2.1. Survey Design

The survey design is a cross-sectional, observational survey among convenience samples of women in low-income, urban settings across three countries. Three distinct survey sites across three continents were chosen; these include Indonesia in Asia, Mexico in North America, and South Africa in Africa. A detailed description of survey areas and dietary patterns for each site is provided elsewhere [6].

In the interest of having a somewhat homogenous population sample with comparable nutrient requirements, only non-pregnant and non-lactating women aged 18–39 years old living in poor, urban, or semi-urban communities were included in the survey. To allow comparison in the diets of women with distinctly different energy requirements, only women with normal weight (BMI 18.5–25 kg/m^2^) or obese women (BMI > 30 kg/m^2^) were included.

### 2.2. Data Collection Procedures

Existing data meeting the study inclusion criteria were already available for South Africa; therefore, no additional fieldwork was undertaken. Data collected for a study that aimed to assess the vitamin A and anthropometric status of South African preschool children and their mothers from four areas with known distinct eating patterns were identified as suitable for the aims of the current survey. Data were collected between June and November 2011, and findings for preschool children were published in 2015 [7]. In Mexico and Indonesia, data were collected prospectively primarily for this analysis. In Indonesia, data were collected in October 2018. In Mexico, data were collected in October–December 2018.

### 2.3. Recruitment of Respondents

In each setting, we aimed to recruit 50 normal-weight and 50 obese women. Criteria for inclusion comprised willingness to participate in the survey and the ability to speak the main local language (Bahasa in Indonesia, Spanish in Mexico, and Afrikaans or English in South Africa). Criteria for exclusion comprised migrants, as well as individuals suffering from cardiovascular, respiratory, endocrine, blood system, gastrointestinal tract, or other systemic diseases.

In Indonesia, participants were recruited from community centers, mainly from local health posts (Posyandu) and from village lists. A total of 192 women were screened for BMI, and 101 (53%) agreed to participate in the study. No women were excluded after analysis. The final sample included 51 normal-weight and 50 obese women.

In Mexico, women who assisted information meetings in various community/health centers in four communities were recruited. These women typically met once a month as part of the activities related to a national health program called “Prospera”, a cash-transfer program. Initially, 270 women were screened for BMI, and only women meeting the inclusion criteria were invited to participate in the study. A total of 138 (51%) women agreed to participate and signed the informed consent form. After data collection, a further 43 women were excluded from the analysis; 38 women did not meet the inclusion criteria for BMI, 1 woman was pregnant, 1 woman was lactating, and 3 women did not complete the interview. The final sample included 47 normal-weight and 48 obese women.

In South Africa, participants were recruited through house-to-house visits. For the current study, a subsample of 46 normal-weight women and 48 obese meeting the inclusion criteria were randomly selected.

### 2.4. Data Collection Procedures in the Field

Data were collected by trained enumerators through face-to-face interviews in the local language using structured survey tools on paper. As previously described [6], the following data were collected: basic socio-demographic characteristics, poverty probability index (only in Indonesia and Mexico) [8], anthropometry measurements, and a quantitative 24 h dietary recall using the multi-pass method. The poverty probability index is a poverty measurement tool based on the answers to 10 country-specific questions about a household’s characteristics and asset ownership and is used to calculate the likelihood that the household is living below the poverty line.

### 2.5. Data Analysis

Descriptive statistics are presented for socio-demographic characteristics and the observed diets by survey area for normal-weight and obese women separately. We first present median (25–75th percentiles) daily intakes of energy, macronutrients, and selected vitamins and minerals of the observed diets. We use these estimated intakes to assess the adequacy of the diet as the proportion of women with intakes below the Estimated Average Requirements (EAR) (i.e., without considering energy intakes) [9]. We then present median (25–75th percentiles) densities of macronutrients and selected vitamins and minerals of the observed diets. The ‘nutrient densities’ were computed as nutrient intakes divided by the energy intake and expressed per 1000 kcal. The ‘critical nutrient density’ was calculated as nutrient requirement (units)/daily energy requirement (kcal) using the WHO Recommended Nutrient Intakes (RNI) [10] and average energy requirement calculated using equations recommended by the FAO/WHO/UNU [11]. For the energy requirement calculations, women were assumed to be from 30 to 59.9 years old, have a sedentary lifestyle, mean height of 158 cm, and mean body weight was 55 kg for normal-weight women and 85 kg for obese women. The resultant reference energy requirement was 1850 kcal for normal-weight women and 2250 kcal for obese women. We use these values to compute the density gap or density excess of the diets; these are calculated as the difference between the ‘observed nutrient density’ and the ‘critical nutrient density’. The observed nutrient density refers to the ratio of nutrient intakes to energy intake, expressed per 1000 kcal, estimated using 24 h dietary recalls. Finally, we present the observed micronutrient density as a percentage of ‘critical nutrient density’ to quantify the extent of these gaps and identify problem nutrients.

Context-specific food composition databases were used in each setting [12,13,14]; in addition, the USA food composition database was used to complete missing nutrient values [15]. The 2004 WHO/FAO Recommended Nutrient Intakes (RNI) [10] and the derived Estimated Average Requirements (EAR) [9] were used as reference values for requirements. The assumed bioavailability was 10% for iron and medium for zinc. The cut-point method (the percentage of individuals in the group with usual intakes below the corresponding EAR) was used to compute the proportion of women at risk of nutrient inadequacy for folate; thiamine; riboflavin; niacin; vitamins B6, B12, C, A, and D; calcium; and zinc.

Statistical analyses were conducted using IBM SPSS Statistic Version 21.0 (IBM Corp., Armonk, NY, USA). Chi-square tests or Mann–Whitney U tests were used to test differences in socio-demographic characteristics between normal-weight and obese women in each survey area. Mann–Whitney U tests were used to test differences in estimated nutrient intakes and the nutrient density of the diet between normal-weight and obese women in each survey area, as well as for the pooled sample.

## 3. Results

### 3.1. Description of Study Respondents

Characteristics of the participants by survey area for normal-weight and obese women are shown in Table 1. The age of the respondents ranged between 18 and 39 years. Obese women were significantly older than normal-weight women (*p* < 0.05). In all settings, normal-weight women had an average BMI of around 22 kg/m^2^, whereas obese women had a mean BMI of 34.0 ± 3.6, 33.9 ± 4.1, and 36.8 ± 7.2 kg/m^2^ in the Indonesian, Mexican, and South African samples, respectively. In Indonesia and Mexico, the vast majority of women were married, whereas in South Africa, the majority were not married. In Indonesia and Mexico (data not available for the South African sample), the likelihood that a household is living below poverty lines did not differ between normal-weight and obese women (*p* > 0.05).

### 3.2. Estimated Daily Intakes Based on a Single 24 h Dietary Recall

The median estimated energy intakes, based on a single 24 h dietary recall, were 1827, 1407, and 1878 kcal for the Indonesian, Mexican, and South African samples, respectively. Although obese women have higher energy requirements than normal-weight women, their estimated energy intakes were not significantly higher (*p* > 0.05). Estimated average intakes are well below daily energy requirements for the respondents, except for normal-weight women in the Indonesian and South African samples. No significant differences in energy and nutrient intakes were observed between normal-weight and obese women for any of the three samples (*p* > 0.05) (shown in Table 2).

### 3.3. Nutrient Adequacy of the Observed Diet Using the Cut-Point Method

The cut-point method (% of EAR) was used to compute the probability of nutrient adequacy for key nutrients (shown in Table 3). The proportion of normal-weight and obese women with nutrient intakes below the EAR was particularly high (i.e., above 50%) for vitamin C and calcium across all three samples. The probability of inadequate intakes was also high for both normal-weight and obese women for all other micronutrients, except zinc in Mexico and folate and vitamin D in South Africa.

### 3.4. Estimated Nutrient Densities Based on a Single 24 h Dietary Recall

The median nutrient densities of nutrients in the diets of normal-weight and obese women are presented by survey area in Table 4. In Indonesia, there were significant differences between normal-weight and obese women in the densities of macronutrients. The densities of protein and saturated fat were higher among normal-weight women, whereas the densities of carbohydrates and fiber were higher among obese women. Furthermore, the nutrient density of vitamin C was significantly higher among obese women when compared to normal women.

In Mexico, there were no significant differences in the nutrient densities of the diets of normal-weight and obese women for any macro- or micronutrients.

In South Africa, there were significant differences between normal-weight and obese women in the densities of macronutrients. The densities of fat and saturated fat were higher among obese women, whereas the density of carbohydrates was higher among normal-weight women. There were no significant differences in the nutrient densities of normal-weight and obese women for any micronutrients.

No significant differences in nutrient densities between the diets of normal-weight and obese women were observed for the pooled sample (three country samples) (*p* > 0.05).

### 3.5. Nutrient Densities of the Diet in Relation to the Critical Nutrient Density

Table 4 also presents calculated critical nutrient densities for normal-weight and obese women. Because the reference energy requirements for normal-weight women are lower than for obese women (1850 kcal vs. 2250 kcal), the critical densities for vitamins and minerals are higher for normal-weight women.

The difference between the observed nutrient densities and the critical nutrient densities presented in Table 4 are presented in Table 5. The table footnotes indicate whether the cells represent density excesses or gaps. Gaps represent nutrients for which intakes would be inadequate even if energy requirements were met. In Indonesia, gaps were identified for thiamine, vitamins C and A, calcium, and iron. In Mexico, gaps were identified for folate; riboflavin; niacin; vitamins B6, B12, and A; calcium; and iron. In South Africa, gaps were identified for folate; vitamins B12, C, and D; calcium; and iron. For various nutrients, the density gap was greater for normal-weight women than for obese women.

The observed micronutrient densities as a percentage of critical nutrient densities presented in Table 4 are presented in Table 6. The cells highlighted in orange indicate a percentage below 80%; these nutrients are classified as “problem nutrients” in the diets of the sampled women. In Indonesia, vitamins C and A were identified as problem nutrients among normal-weight women; calcium and iron were identified as problem nutrients for both normal-weight and obese women. In Mexico, calcium was identified as a problem nutrient among normal-weight women; folate, vitamins B6 and B12, and iron were identified as problem nutrients for both normal-weight and obese women. In South Africa, folate, vitamin D, calcium, and iron were identified as problem nutrients for both normal-weight and obese women.

## 4. Discussion

The nutrient density principle seeks to ensure that nutrients are provided in sufficient concentrations in the diet to satisfy individuals’ nutrient needs if they consume sufficient food to maintain energy balance. When there is an apparent energy deficit, some nutrient requirements can be met by consuming more of the same foods (i.e., existing diet), whereas other nutrient deficits would require dietary modifications (i.e., more nutrient-dense dietary sources). This current study aimed to identify which nutrients would require dietary modifications among normal-weight and obese women across different settings. This work provided an opportunity to develop a rigorous approach towards nutrient-density analysis, allowing the assessment of specific nutrient inadequacies in various settings to provide valuable information for efforts to tackle deficiencies.

The more calories one consumes from a given diet, and calorie intakes can vary widely, the more macro- and micronutrients one can consume, contributing to achieving or surpassing one’s recommended intake level for these two classes of nutrients. For example, marathon swimmers can consume up to 5000 kcal in a training day, whereas individuals on a weight-loss diet can consume 1000 kcal, but both would require a common RNI of vitamin C of 45 mg/day. The critical nutrient density for this vitamin of the former would be 9 mg/1000 kcal and the latter, 45 mg/1000 kcal.

The hypothesis and conceptual framework of this study, with a comparison of women of normal vs. obese weight status in three settings, was that the groups would differ in daily energy consumption. We expected that those with a BMI > 30 kg/m^2^ would consume more calories daily than the group with a BMI of 18.5–24.9 kg/m^2^. This relationship is obvious in the active phase of weight gain into obesity, but even in stable-weight obesity status, the energy cost of supporting and transporting additional body mass and the demands of compensatory musculature would predict a differential caloric consumption to achieve energy balance. As seen in Table 4, the UN agency consultancy guides us to a 22% higher energy expenditure (1850 vs. 2250 kcal/day) as the normative reference for normal-weight and obesity [11].

The lack of clear difference in energy intake between normal-weight and obese is interesting but in no way unprecedented. Empirical evidence for the difference from dietary intake studies is widely lacking, but under-reporting of food intake has been associated with having higher body weight [16,17,18]. Even with the accuracy of doubly labeled water estimation for long-term energy expenditure, there is no consistency or consensus on differential energy demand with obesity [19,20,21]. As such, our present findings, in three distinct nations, are not atypical for obesity–normal-weight comparisons. Under-reporting is a common error in 24 h dietary data; as such, interpretation of findings requires caution.

In this study, survey procedures were standardized across countries; however, cross-country comparisons are limited due to the lack of representativeness of the small convenience samples. A further limitation of the study is that it was only feasible to conduct a single 24 h dietary recall for each respondent; as such, we were not able to account for day-to-day variability of intakes. Additionally, country-specific food composition databases were used in each setting, and the databases in Indonesia and Mexico included a limited selection of micronutrients. Nevertheless, within each setting, with unique cuisines and dietary habits, similar contrasts between normal-weight and obese women were explored.

As shown in the first data row of Table 2, although the reported energy intakes for normal-weight women in Indonesia and South Africa tightly approximate the 1850 kcal/day assumption of the WHO/FAO/UNU [11], the other four groups report daily caloric intake far inferior to the corresponding reference value. It behooves us to examine some of the practical implications and consequences of these findings for critical nutrient density applications. For the normal-weight women of Indonesia, we used 1850 kcal reference in the denominator, and the critical vitamin C density 24.4 mg/1000, as compared to 2250 kcal for their obese peers, for a density calculation of 20.0 mg/1000 kcal (based on WHO RNI’s [10]). However, in fact, with the pooled average of 1844 kcal for both weight groups—and a vitamin C RNI of 45 mg—the peri-urban Indonesian women need 24.3 mg/1000 kcal density in their diet. If other settings have no differentiation in energy intake across the BMI spectrum, one can use a universal denominator for the given age–sex group.

As a consequence of the application of WHO/FAO/UNU references, the density-gap data in Table 5 are generally valid in two instances (normal weight in Indonesia and South Africa) but are variously under-estimated for the remaining four groups.

## 5. Conclusions

We conclude that to apply a true critical nutrient density reference for a population or subgroup, it may often require some approximate estimate of the habitual energy intake from an empirical source and not just from theoretical estimates from the WHO/FAO/UNU. It was a discovery in the analysis of the present data that revealed the inconsistencies of prediction by conventional energy references. Because the energy-based (denominator value) is smaller, this will logically signify that there would be more “problem nutrients” and larger nutrient gaps in the diets examined with this nutrient density approach, and efforts of improved food selection or food- or biofortification will frequently be indicated.

## Figures and Tables

**Table 1 nutrients-13-04016-t001:** Characteristics of the participants by survey area for normal-weight vs. obese women ^1^.

		West Java, Indonesia		Querétaro, Mexico		Western Cape, South Africa	
		Normal-Weight ^2^ (*n* = 51)	Obese ^3^ (*n* = 50)	Normal-Weight (*n* = 47)	Obese (*n* = 48)	Normal-Weight (*n* = 46)	Obese (*n* = 48)
Age (years), mean ± SD		28.5 ± 5.8 *	30.2 ± 6.3	28.9 ± 4.4 *	31.1 ± 4.0	26.2 ± 4.8 *	30.4 ± 5.9
Age	18–24 years, *n* (%)	14 (27%) *	11 (23%) *	10 (23%) *	5 (10%)	23 (50%) *	9 (19%)
	25–31 years, *n* (%)	20 (39%)	18 (36%)	20 (43%)	14 (29%)	14 (30%)	18 (38%)
	32–39 years, *n* (%)	17 (33%)	22 (44%)	16 (34%)	29 (60%)	9 (20%)	21 (44%)
Body weight (kg), mean ± SD		52.0 ± 5.6 *	77.5 ± 10.8	55.2 ± 7.1 *	80.4 ± 12.5	52.6 ± 6.7 *	91.2 ± 16.9
Height (cm), mean ± SD		154.4 ± 4.8	150.8 ± 4.8	157.2 ± 5.6	153.7 ± 5.8	156.5 ± 5.6	157.5 ± 6.5
BMI (kg/m^2^), mean ± SD		21.8 ± 1.7 *	34.0 ± 3.6	22.3 ± 2.3 *	33.9 ± 4.1 *	21.4 ± 2.1	36.8 ± 7.2
Marital status	Married, *n* (%)	38 (75%) *	42 (84 %)	32 (68%)	32 (67%)	10 (22%) *	19 (40%)
	Not married, *n* (%)	13 (25%)	8 (16 %)	15 (32%)	16 (33%)	36 (78%)	29 (60%)
Poverty Probability Index ^4^	$1.90 2011 PPP (%), mean ± SD	0.4 ± 0.8	0.5 ± 1.0	8.2 ± 4.4	8.2 ± 5.6	-	-
	$3.10 2011 PPP (%), mean ± SD	7.7 ± 10.4	9.9 ± 11.8	22.0 ± 12.4	21.3 ± 12.9	-	-
	$1.90 2011 PPP (%), median (25–75th percentile)	0 (0–0.2)	0.1 (0–0.6)	8.3 (4.3–11.9)	7.0 (4.8–10.4)	-	-
	$3.10 2011 PPP (%), median (25–75th percentile)	2.4 (1.1–7.8) *	4.1 (2.4–14.3)	21.8 (10.5–33.2)	19.4 (13.9–26.8)	-	-

PPP—Purchasing power parity; SD—standard deviation. ^1^ Significant differences between normal-weight and obese women within each sample are indicated with an *; ^2^ normal weight is defined as body mass index (BMI) 18.5–25 kg/m^2^; ^3^ Obese is defined as BMI ≥ 30 kg/m^2^; ^4^ Likelihood that a household is living below national and international poverty lines.

**Table 2 nutrients-13-04016-t002:** Median estimated daily intakes based on a single 24 h dietary recall by survey area for normal-weight vs. obese women.

	Daily Requirements		Median Estimated Daily Intakes Per Day											
			West Java, Indonesia				Querétaro, Mexico				Western Cape, South Africa			
	RNI ^1^	EAR ^2^	Normal-Weight ^3^ (*n* = 51)		Obese ^4^ (*n* = 50)		Normal-Weight (*n* = 47)		Obese (*n* = 48)		Normal-Weight (*n* = 46)		Obese (*n* = 48)	
			Median	25–75 Percentile	Median	25–75 Percentile	Median	25–75 Percentile	Median	25–75 Percentile	Median	25–75 Percentile	Median	25–75 Percentile
Energy (kcal)			1820	(1401–2156)	1864	(1336–2203)	1480	(1169–1966)	1369	(966–1934)	1878	(1311–2432)	1857	(1494–2345)
Protein (g)	-	-	59	(47–72)	47	(36–71)	51	(37–67)	50	(33–72)	65	(49–93)	66	(52–83)
Fat (g)	-	-	54	(39–69)	56	(37–68)	53	(27–66)	43	(33–67)	69	(42–84)	79	(56–103)
Saturated fat (g)	-	-	13	(10–21)	11	(8–15)	9	(5–20)	10	(6–15)	20	(14–29)	26	(17–37)
Mono-unsaturated fat (g)	-	-	13	(9–19)	12	(8–19)	8	(5–17)	11	(6–17)	21	(14–30)	27	(17–35)
Poly-unsaturated fat (g)	-	-	6	(5–9)	7	(4–10)	6	(3–11)	7	(3–10)	17	(12–25)	18	(12–22)
Cholesterol (mg)	-	-	-	-	-	-	127	(86–185)	91	(48–225)	201	(118–415)	258	(124–520)
Carbohydrates (g)	-	-	247	(187–320)	288	(188–379)	214	(154–263)	189	(140–251)	256	(168–309)	218	(181–266)
Fiber (g)	-	-	5.5	(3.7–8.2)	6.6	(4.5–12.0)	11	(7–18)	8	(5–19)	14	(8–20)	13	(10–20)
Vitamins														
Folate DFE (μg DFE)	400	320	-	-	-		92.3	(46.0–190.7)	66.0	(27.9–109.6)	229	(155–378)	266	(196–376)
Thiamine (mg)	1.1	0.9	0.9	(0.7–1.3)	0.9	(0.6–1.4)	0.8	(0.5–1.4)	0.7	(0.6–1.1)	1.1	(0.7–1.6)	1.2	(0.9–1.7)
Riboflavin (mg)	1.1	0.9	1.1	(0.7–1.5)	0.9	(0.6–1.5)	0.7	(0.5–1.5)	0.7	(0.5–1.2)	1.1	(0.7–1.9)	1.1	(0.8–1.7)
Niacin (mg NE)	14	11	15.8	(11.8–21.3)	15.6	(10.1–22.3)	10.6	(5.2–16.6)	9.6	(5.4–18.3)	20.2	(14.7–29.0)	21.4	(17.4–27.0)
Vitamin B6 (mg)	1.3	1.1	-	-	-	-	0.6	(0.3–1.3)	0.6	(0.2–1.6)	3.8	(2.2–5.0)	3.5	(2.9–4.9)
Vitamin B12 (μg)	2.4	2.0	-	-	-	-	1.6	(0.6–2.6)	1.1	(0.5–2.0)	2.0	(0.8–4.1)	2.9	(1.9–4.0)
Vitamin C (mg)	45	37	14.2	(4.5–34.7)	30.1	(11.2–68.8)	34.8	(12.9–83.8)	30.5	(11.6–66.2)	26.6	(11.5–56.5)	22.5	(11.0–78.3)
Vitamin A (c)	500	357	375	(241–586)	369	(209–547)	271	(154–625)	245	(114–454)	416	(241–704)	429	(324–745)
Vitamin D (μg)	5	5	-	-	-	-	-	-	-	-	2.4	(1.2–8.6)	3.0	(1.4–8.6)
Minerals														
Calcium (mg)	1000	833	653	(368–891)	481	(281–805)	631	(403–868)	610	(341–924)	247	(128–485)	271	(171–400)
Iron (mg)	29.4 ^5^	-	14.5	(10.9–20.0)	13.4	(8.6–19.2)	8.8	(6.0–14.5)	9.1	(6.2–12.7)	11.2	(7.4–15.9)	11.9	(9.6–15.4)
Zinc (mg)	4.9 ^6^	4.1	6.4	(4.7–8.2)	5.7	(4.1–8.7)	4.6	(2.4–7.4)	4.1	(1.9–7.0)	10.5	(7.3–15.4)	11.7	(9.4–14.7)
Phosphorus (mg)	-	-	778	(567–1132)	707	(505–969)	-	-	-	-	863	(564–1187)	789	(596–1081)
Copper (mg)	-	-	1.2	(0.8–2.4)	1.1	(0.7–1.9)	-	-	-	-	0.9	(0.6–1.1)	1.0	(0.7–1.2)
Potassium (mg)	-	-	768	(522–1278)	850	(561–1640)	1260	(917–1609)	1109	(617–1522)	1809	(1278–2639)	1823	(1502–2223)
Sodium (mg)	-	-	773	(358–1468)	709	(377–1198)	1426	(897–2130)	1486	(747–2132)	1800	(1012–3070)	1855	(1330–2969)

DFE—dietary folate equivalent; NE—niacin equivalents; RE—retinol equivalents; TE—tocopherol equivalents. ^1^ Recommended nutrient intakes (RNIs) for non-pregnant, non-lactating women aged 19–50 years [10]. ^2^ Estimated average requirements (EAR) for non-pregnant, non-lactating women aged 19–50 years, calculate values based on FAO/WHO RNIs [9]. ^3^ Normal weight is defined as body mass index (BMI) 18.5–25 kg/m^2^. ^4^ Obese is defined as BMI ≥ 30 kg/m^2^. ^5^ Assuming 10% bioavailability. ^6^ Assuming medium bioavailability.

**Table 3 nutrients-13-04016-t003:** Population prevalence of inadequate intakes based on a single 24 h dietary recall by survey area for normal-weight vs. obese women.

	Estimated Average Requirements (EAR) ^1^	Proportion of Women with Intakes below the EAR (%)					
		West Java, Indonesia		Querétaro, Mexico		Western Cape, South Africa	
		Normal-Weight ^2^ (*n* = 51)	Obese ^3^ (*n* = 50)	Normal-Weight (*n* = 47)	Obese (*n* = 48)	Normal-Weight (*n* = 46)	Obese (*n* = 48)
Folate DFE (μg DFE)	320	-	-	91 ^4^	94 ^4^	67 ^4^	69 ^4^
Thiamine (mg)	0.9	53 ^4^	46	57 ^4^	67 ^4^	37	23
Riboflavin (mg)	0.9	43	46	62 ^4^	58 ^4^	35	40
Niacin (mg NE)	11	20	30	53 ^4^	65 ^4^	11	2
Vitamin B6 (mg)	1.1	-	-	70 ^4^	69 ^4^	9	4
Vitamin B12 (μg)	2.0	-	-	66 ^4^	75 ^4^	50	27
Vitamin C (mg)	37	78 ^4^	58 ^4^	51 ^4^	56 ^4^	61 ^4^	60 ^4^
Vitamin A (μg RE)	357	45	46	55 ^4^	67 ^4^	41	38
Vitamin D (μg)	5	-	-	-	-	70 ^4^	58 ^4^
Calcium (mg)	833	73 ^4^	76 ^4^	72 ^4^	67 ^4^	96 ^4^	96 ^4^
Zinc (mg)	4.1	18	24	47	50	7	0

DFE—dietary folate equivalent; NE—niacin equivalents; RE—retinol equivalents. ^1^ Estimated average requirements (EAR) for non-pregnant, non-lactating women aged 19–50 years, calculate values based on FAO/WHO RNIs [9]. ^2^ Normal weight is defined as body mass index (BMI) 18.5–25 kg/m^2^. ^3^ Obese is defined as BMI ≥ 30 kg/m^2^. ^4^ The proportion of women with intakes below the EAR is above 50%.

**Table 4 nutrients-13-04016-t004:** Median density of the diet based on a single 24 h dietary recall by survey area for normal-weight vs. obese women.

	Critical Nutrient Density, Unit/1000 kcal ^1^		Nutrient Density of the Diet, per 1000 kcal											
	Normal-Weight Women	Obese Women	West Java, Indonesia				Querétaro, Mexico				Western Cape, South Africa			
			Normal-Weight ^2^ (*n* = 51)		Obese ^3^ (*n* = 50)		Normal-Weight (*n* = 47)		Obese (*n* = 48)		Normal-Weight (*n* = 46)		Obese (*n* = 48)	
			Median	25–75 Percentiles	Median	25–75 Percentiles	Median	25–75 Percentiles	Median	25–75 Percentiles	Median	25–75 Percentiles	Median	25–75 Percentiles
Macronutrients														
Protein (g/1000 kcal)			33.5	(29.1–37.9)	27.7 ^4^	(23.7–35.6)	34.9	(25.5–41.7)	35.9	(28.1–47.1)	33.8	(29.1–42.4)	35.2	(29.6–42.1)
Fat (g/1000 kcal)	-	-	31.1	(26.8–37.2)	28.9	(21.8–36.7)	33.4	(25.9–41.8)	33.0	(26.2–42.4)	35.8	(29.8–42.5)	41.4 ^4^	(33.3–47.4)
Saturated fat (g/1000 kcal)	-	-	8.1	(5.6–11.1)	6.5 ^4^	(3.9–9.8)	7.7	(3.9–11.1)	7.2	(5.2–10.6)	11.5	(8.6–14.3)	14.0 ^4^	(10.7–16.5)
Cholesterol (mg/1000 kcal)	-	-	-	-	-	-	83.2	(58.3–120.0)	70.6	(40.0–127.4)	96.9	(65.8–215.5)	124.7	(75.3–242.5)
Carbohydrates (g/1000 kcal)	-	-	145.7	(124.0–160.1)	153.4 ^4^	(132.5–177.5)	144.8	(119.2–159.6)	137.2	(114.3–154.9)	132.3	(114.8–147.6)	120.9 ^4^	(104.4–133.2)
Fiber (g/1000 kcal)	-	-	3.2	(2.0–4.9)	3.9 ^4^	(2.9–5.6)	7.9	(4.6–11.6)	6.2	(3.3–12.2)	7.4	(5.4–9.7)	7.7	(5.9–9.3)
Vitamins														
Folate DFE (μg DFE/1000 kcal)	216.2	177.8	-	-	-	-	59.9	(34.7–130.6)	48.5	(20.6–90.1)	137.1	(98.3–166.5)	137.8	(103.5–197.1)
Thiamine (mg/1000 kcal)	0.59	0.49	0.5	(0.4–0.7)	0.5	(0.4–0.7)	0.6	(0.5–0.8)	0.6	(0.4–0.7)	0.6	(0.5–0.8)	0.6	(0.5–0.8)
Riboflavin (mg/1000 kcal)	0.59	0.49	0.6	(0.4–0.8)	0.5	(0.4–0.8)	0.5	(0.4–0.8)	0.5	(0.4–0.7)	0.6	(0.4–0.9)	0.6	(0.4–1.0)
Niacin (mg NE/1000 kcal)	7.6	6.2	9.0	(7.3–10.9)	8.6	(7.1–10.5)	7.3	(3.5–11.4)	7.5	(4.3–10.7)	11.8	(9.1–13.9)	11.7	(9.2–14.8)
Vitamin B6 (mg/1000 kcal)	0.70	0.58	-	-	-	-	0.5	(0.3–0.8)	0.4	(0.2–1.0)	2.1	(1.5–2.5)	1.9	(1.4–2.7)
Vitamin B12 (μg/1000 kcal)	1.3	1.1	-	-	-	-	0.9	(0.5–2.1)	0.8	(0.4–1.4)	1.2	(0.6–2.5)	1.4	(1.2–2.4)
Vitamin C (mg/1000 kcal)	24.3	20.0	8.3	(1.8–19.7)	18.4 ^4^	(9.1–30.3)	24.6	(7.7–51.5)	21.6	(8.3–46.8)	16.8	(5.8–29.1)	15.8	(6.8–41.4)
Vitamin A (μg RE/1000 kcal)	270.3	222.2	211.9	(145.7–343.8)	195.7	(147.2–299.6)	226.3	(95.0–389.7)	196.4	(81.5–303.8)	232.9	(140.8–323.0)	251.1	(184.3–368.1)
Vitamin D (μg/1000 kcal)	2.7	2.2	-	-	-	-	-	-	-	-	1.2	(0.73.8)	1.5	(0.7–4.0)
Minerals														
Calcium (mg/1000 kcal)	540.5	444.4	377.8	(238.6–482.2)	267.0	(210.2–396.3)	422.0	(298.7–561.5)	398.0	(289.1–507.7)	115.4	(78.2–233.6)	130.5	(98.1–179.0)
Iron (mg/1000 kcal)	15.9 ^5^	13.1	8.0	(6.6–11.0)	7.6	(6.2–10.0)	6.6	(4.4–7.8)	6.2	(4.9–8.0)	6.5	(5.4–7.5)	6.2	(5.4–7.7)
Zinc (mg/1000 kcal)	2.65 ^6^	2.18	3.4	(2.9–4.5)	3.3	(2.6–4.1)	2.7	(1.8–4.8)	2.8	(2.0–4.4)	5.5	(4.6–6.9)	6.3	(5.1–7.9)
Phosphorus (mg/1000 kcal)	-	-	468.5	(392.9–538.6)	394.5	(317.7–518.4)					435.1	(381.7–540.4)	426.4	(389.6–526.4)
Copper (mg/1000 kcal)	-	-	0.7	(0.5–1.0)	0.7	(0.4–0.9)					0.5	(0.4–0.5)	0.5	(0.4–0.6)
Potassium (mg/1000 kcal)	-	-	439.5	(340.4–616.1)	528.8	(398.9–750.2)	906.5	(662.6–1120.1)	853.4	(508.2–1221.7)	953.2	(770.6–1215.5)	968.5	(805.8–1128.9)
Sodium (mg/1000 kcal)	-	-	465.3	(246.9–804.9)	464.7	(221.7–709.5)	1059.0	(633.8–1378.3)	1200.5	(507.2–1548.6)	1018.3	(663.0–1247.5)	1057.5	(804.8–1425.4)

DFE—dietary folate equivalent; NE—niacin equivalents; RE—retinol equivalents; TE—tocopherol equivalents. ^1^ Calculated as nutrient requirement (units)/daily energy requirement (kcal), using the WHO recommended nutrient intakes (RNI) [10] and the energy requirement of 1850 kcal for normal-weight women and 2250 kcal for obese women [11]. ^2^ Normal weight is defined as body mass index (BMI) 18.5–25 kg/m^2^. ^3^ Obese is defined as BMI ≥ 30 kg/m^2^. ^4^ The estimated daily intake was significantly different between normal weight and obese women (Mann–Whitney U test, *p* < 0.05). ^5^ Assuming 10% bioavailability. ^6^ Assuming medium bioavailability.

**Table 5 nutrients-13-04016-t005:** Micronutrient density gap or density excess of the diets, in units per 1000 kcal, by survey area for normal-weight vs. obese women.

	Density Gap or Excess of the Diet, Per 1000 kcal ^1^					
	West Java, Indonesia		Querétaro, Mexico		Western Cape, South Africa	
	Normal-Weight ^2^ (*n* = 51)	Obese ^3^ (*n* = 50)	Normal-Weight (*n* = 47)	Obese (*n* = 48)	Normal-Weight (*n* = 46)	Obese (*n* = 48)
Folate DFE (μg DFE/1000 kcal)	-	-	−156.3 ^4^	−129.3 ^4^	−79.1 ^4^	−40 ^4^
Thiamine (mg/1000 kcal)	−0.09 ^4^	0.01	0.01	0.11	0.01	0.11
Riboflavin (mg/1000 kcal)	0.01	0.01	−0.09 ^4^	0.01	0.01	0.11
Niacin (mg NE/1000 kcal)	1.4	2.4	−0.3 ^4^	1.3	4.2	5.5
Vitamin B6 (mg/1000 kcal)	-	-	−0.2v	−0.18 ^4^	1.4	1.32
Vitamin B12 (μg/1000 kcal)	-	-	−0.4 ^4^	−0.3 ^4^	−0.1 ^4^	0.3
Vitamin C (mg/1000 kcal)	−16 ^4^	−1.6 ^4^	0.3	1.6	−7.5 ^4^	−4.2 ^4^
Vitamin A (μg RE/1000 kcal)	−58.4 ^4^	−26.5	−44 ^4^	−25.8 ^4^	−37.4 ^4^	28.9
Vitamin D (μg/1000 kcal)	-	-	-	-	−1.5 ^4^	−0.7 ^4^
Calcium (mg/1000 kcal)	−162.7 ^4^	−177.4 ^4^	−118.5 ^4^	−46.4 ^4^	−425.1 ^4^	−313.9 ^4^
Iron (mg/1000 kcal)	−7.9 ^4^	−5.5 ^4^	−9.3 ^4^	−6.9 ^4^	−9.4 ^4^	−6.9 ^4^
Zinc (mg/1000 kcal)	0.75	1.12	0.05	0.62	2.85	4.12

^1^ Calculated as ‘Nutrient density of the diet’—‘Critical nutrient density’. The critical nutrient density was calculated as nutrient requirement (units)/daily energy requirement (kcal), using the WHO recommended nutrient intakes (RNI) [10] and the energy requirement of 1850 kcal for normal-weight women and 2250 kcal for obese women [11]. ^2^ Normal weight is defined as body mass index (BMI) 18.5–25 kg/m^2^. ^3^ Obese is defined as BMI ≥ 30 kg/m^2^. ^4^ These cells represent a density gap.

**Table 6 nutrients-13-04016-t006:** Observed micronutrient density as a percentage of critical nutrient density by survey area for normal-weight vs. obese women.

	Observed Nutrient Density as a Percentage of Critical Nutrient Density (%) ^1^					
	West Java, Indonesia		Querétaro, Mexico		Western Cape, South Africa	
	Normal-Weight ^2^ (*n* = 51)	Obese ^3^ (*n* = 50)	Normal-Weight (*n* = 47)	Obese (*n* = 48)	Normal-Weight (*n* = 46)	Obese (*n* = 48)
Folate DFE (μg DFE/1000 kcal)	-	-	28% ^4^	27% ^4^	63% ^4^	78% ^4^
Thiamine (mg/1000 kcal)	85% ^5^	102% ^6^	102% ^6^	122% ^6^	102% ^6^	122% ^6^
Riboflavin (mg/1000 kcal)	102% ^6^	102% ^6^	85% ^5^	102% ^6^	102% ^6^	122% ^6^
Niacin (mg NE/1000 kcal)	118% ^6^	139% ^6^	96% ^5^	121% ^6^	155% ^6^	189% ^6^
Vitamin B6 (mg/1000 kcal)	-	-	71% ^4^	69% ^4^	300% ^6^	328% ^6^
Vitamin B12 (μg/1000 kcal)	-	-	69% ^4^	73% ^4^	92% ^5^	127% ^6^
Vitamin C (mg/1000 kcal)	34% ^4^	92% ^5^	101% ^6^	108% ^6^	69% ^4^	79% ^4^
Vitamin A (μg RE/1000 kcal)	78% ^4^	88% ^5^	84% ^5^	88% ^5^	86% ^5^	113% ^6^
Vitamin D (μg/1000 kcal)	-	-	-		44% ^4^	68% ^4^
Calcium (mg/1000 kcal)	70% ^4^	60% ^4^	78% ^4^	90% ^5^	21% ^4^	29% ^4^
Iron (mg/1000 kcal)	50% ^4^	58% ^4^	42% ^4^	47% ^4^	41% ^4^	47% ^4^
Zinc (mg/1000 kcal)	128% ^6^	151% ^6^	102% ^6^	128% ^6^	208% ^6^	289% ^6^

^1^ The ‘critical nutrient density’ was calculated as nutrient requirement (units)/daily energy requirement (kcal) using the WHO recommended nutrient intakes (RNI) [10] and the energy requirement of 1850 kcal for normal-weight women and 2250 kcal for obese women [11]. ^2^ Normal weight is defined as body mass index (BMI) 18.5–25 kg/m^2^. ^3^ Obese is defined as BMI ≥ 30 kg/m^2^. ^4^ These cells indicate a percentage below 80%; classified as “problem nutrients”. ^5^ These cells indicate a percentage between 80% and 100%. In the present paper, we have altered the convention of problem nutrient from meeting 100% of the critical nutrient density to a more conservative 80% or less; as such. these are also defined as “problem nutrients”. ^6^ These cells indicate a percentage above 100%; these intakes are adequate.

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
