# Peer review of "Nutrient Density as a Dimension of Dietary Quality: Findings of the Nutrient Density Approach in a Multi-Center Evaluation"

_nutrients, 2021, doi:10.3390/nu13114016_

Round 1
Reviewer 1 Report
This is an interesting comparative article.
I cannot see any mention of you seeking ethical approval to collect the data in the methods and informed consent is important.
Further detail required in methods:
How might different food composition databases impact on findings. Sample size - no calculation as to whether it is adequate to be detecting all differences. Can you provide more details on poverty index calculation
Also a paragraph on the limitations of the study should be included. Only one 24 hour recall was conducted. There is no mention about the validity of the dietary intake data to begin with such as misreporting. Small sample size.
On a technical note Table 1 column 1 is very difficult to read and needs reformatting.
Author Response
Dear reviewer, we thank you for your careful review. Your comments have helped improve our paper. Please refer to the attached document for our responses. Kindest.

Reviewer 2 Report
Thank you for the opportunity to review this interesting manuscript. The concept of critical nutrient density is potentially useful but is critically dependent on an accurate estimate of the energy requirements of the group to which it is applied. The accuracy of dietary intake reporting is mediocre with under-reporting common – this is a problem for comparing dietary data against absolute nutrient benchmarks. The problem could be partly addressed by estimating the nutrient density of the reported diet (i.e. the nutrient composition of the diet) to adjust for under-reporting if the actual dietary energy requirement could be accurately estimated (and the food intake was uniformly under-reported on a group basis). This requires a lot of assumptions - however the authors here go a step further in using a theoretical energy requirement based on broad groups defined by BMI. I think this approach needs more accurate dietary intake data and data adequate to better estimate true energy requirement.
Minor comments:
The explanation of ‘critical nutrient density is not referenced, so it is assumed that this concept has been developed by the authors and is here being explored for the first time. This could be made clearer (Lines 44-51).
Line 50: it is not clear why obese women (having a lower critical nutrient density because they have a higher energy requirement) have a greater nutrient ‘allowance’ than non-obese women – isn’t it the same? Or is it a greater energy ‘allowance’ for the intake of nutrient?
Line 97-99 : there appears to be a contradiction – 138 women are stated to have met the BMI criteria (line 97), but after data collection 32 of these women were excluded because they did not meet the BMI criteria (line 99). Additionally the final sample size does not match the exclusions – 138 signed consent, 37 were excluded after data collection should total 101. But the final sample was 47 + 48 = 95
Lines 123/124 – the energy requirement for normal weight women and obese women is stated without justification. The definition of the observed nutrient density should be included in this section.
The methods section should outlie why folate, vitamin B6, vitamin B12 and vitamin D intake was not estimated for the Indonesian sample (perhaps around line 135)
Line 129: The authors reference 4 food composition databases for 3 samples. Could they be more explicit about which compositional databases were used for which sample.
Table 6 – at least 2 of the cells highlighted in light green are less than 80%.
Discussion
Under-reporting is clearly a potentially serious problem with this analysis – it is very common to find greater under-reporting among obese people. It is unclear whether the authors accept a greater level of under-reporting or think that the reporting is equally accurate for both normal weight and obese. The serious problem with under-reporting impacting on this analysis is that it can be differential (some foods under-reported to a greater extent) or non-differential (the amount of foods reported to be consumed are less than the amount actually consumed, but all the foods are represented in the correct proportions). The authors are clearly aware of this but it remains a serious issue in interpreting the adequacy of dietary intake data.
If the density gaps information is valid in 2 instances (line 284) examination of table 5 for these 2 instances doesn’t advance much on the more classical analysis for those 2 instances in Table 3. This doesn’t strengthen the argument for use of ‘critical nutrient density’. The reason the authors suggest the two instances to be valid appears to be because the estimated energy intake is close to the theoretical energy requirement they used. I am not as confident that this makes them valid – it would be more comforting if all of the groups were close to the theoretical estimate. I wonder whether the same statement about validity would have been made if the 2 instances were obese groups rather than normal weight groups.
I am left wondering why the authors didn’t use the estimated group mean energy intake to calculate ‘critical nutrient density’ rather than the estimates from FAO/WHO/UNU for the broad groups. If the nutrient requirements are truly independent of energy requirement, this would define the energy space within which the nutrient requirements needed to be achieved. The authors call for an empirical source for energy intake in their conclusion, but don’t explore the source they had. Perhaps they are ambivalent about the accuracy of the energy intake data, but they do not make this clear.
I don’t accept the authors conclusion that ‘there would be more “problem nutrients” and larger nutrient gaps in the diets examined with this nutrient-density approach, and efforts of improved food selection or food- or biofortification will frequently be indicated’ (lines 290-292) because I don’t think they know enough about the errors in their dietary data, and the deficiency in the theoretical energy requirements used. They may have taken a particular view about whether their measured energy intake was closer to the actual energy intake than the theoretical energy requirement, but this is an assumption about which they should be explicit. It doesn’t seem like the only possibility.
Author Response

(The authors gave the same response as above.)

Round 2
Reviewer 2 Report
Thank you for your responses to my original review comments - there are issues I had not previously appreciated and I have learned from your rebuttal.
I am remain confused by modifications to Table 6 - each of the highlighted colours (i.e. all the table entries) define or classify 'problem nutrients' (as stated in the footnotes). Is this correct?